# Morphology and Microwave-Absorbing Performances of Rubber Blends with Multi-Walled Carbon Nanotubes and Molybdenum Disulfide

**DOI:** 10.3390/nano13101644

**Published:** 2023-05-15

**Authors:** Le Huang, Jingru Chen, Bingjun Liu, Pengfei Zhao, Lusheng Liao, Jinlong Tao, Yueqiong Wang, Bingbing Wang, Jing Deng, Yanfang Zhao

**Affiliations:** 1Key Laboratory of Advanced Materials of Tropical Island Resources of Ministry of Education, School of Materials Science and Engineering, Hainan University, Haikou 570228, China; hl1216278916@163.com (L.H.); 18689996081@163.com (J.C.); lbj18089863579@163.com (B.L.); 13138973831@163.com (J.D.); 2Guangdong Provincial Key Laboratory of Natural Rubber Processing, Agricultural Products Processing Research Institute, Chinese Academy of Tropical Agricultural Sciences, Zhanjiang 524001, China; pengfeizhao85ac@163.com (P.Z.); jinlongt1983@163.com (J.T.); yqw215@163.com (Y.W.); wangbbhn@163.com (B.W.); 3Hainan Provincial Key Laboratory of Natural Rubber Processing, Zhanjiang 524001, China

**Keywords:** molybdenum disulfide, multi-walled carbon nanotubes, natural rubber, nitrile butadiene rubber, electromagnetic loss

## Abstract

This study details microwave-absorbing materials made of natural rubber/nitrile butadiene rubber (NR/NBR) blends with multi-walled carbon nanotubes (MWCNTs) and molybdenum disulfide (MoS_2_). The mechanical blending method and the influences of fabrication on the morphology and microwave-absorbing performance of resulting compounds were logically investigated. It was found that interfacial differences between the fillers and matrix promote the formation of MWCNTs and MoS_2_ networks in NR/NBR blends, thus improving microwave-absorbing performance. Compared with direct compounding, masterbatch-based two-step blending is more conducive to forming interpenetrating networks of MWCNTs/MoS_2_, endowing the resulting composite with better microwave attenuation capacity. Composites with MWCNTs in NR and MoS_2_ in NBR demonstrate the best microwave-absorbing performance, with a minimum reflection loss of −44.54 dB and an effective absorption bandwidth of 3.60 GHz. Exploring the relationship between morphology and electromagnetic loss behavior denotes that such improvement results from the selective distribution of dual fillers, inducing networking and multi-component-derived interfacial polarization enhancement.

## 1. Introduction

With the rapid development of modern electronic information technology, mobile communication, wireless electronic equipment, industrial equipment, etc., electromagnetic pollution has become more and more complex and serious [1]. Electromagnetic radiation not only interferes with the normal function of electronic equipment but also threatens human health [2,3]. Developing microwave-absorbing materials (MAMs) that can attenuate electromagnetic waves is one of the most effective methods to solve this problem [4,5]. Owing to their merits of excellent flexibility, low density, corrosion resistance, and easy processing, elastomeric composites derived from rubber and electromagnetic lossy blocks have been considered to be intriguing candidates for the fabrication of light microwave absorbers with significant attenuation over a wide frequency range [6].

Like other microwave-absorbing composites, microwave-absorbing performances of elastomeric composites are stemmed from the electromagnetic loss network throughout the matrix [7]. While the microwave attenuation of dielectric lossy blocks (such as conductive carbon black [8], carbon nanotube [9], graphene [10], etc.) is mainly derived from their high conduction loss, polarization relaxation, high magnetic saturation, and low coercivity dominantly contribute to the high microwave-absorbing performance of magnetic lossy blocks including Fe, Co, Ni, and their oxides [11,12,13]. It is well-accepted that either dielectric blocks or magnetic fillers demonstrate inefficient microwave attenuation. Therefore, over the past decade, extensive efforts have been devoted to investigating the synergetic effects of electromagnetic hybrids on improving the microwave-absorbing performance of elastomeric composites [14,15]. For example, the co-incorporation of carbon black and iron oxide affords natural rubber emulsion with a strong absorption of −42.00 dB at 4.20 GHz when the thickness is 2.20 mm [16]. Due to the synergistic effect between dielectric loss and magnetic loss, silicone rubber composites with iron oxide-anchored reduced graphene oxide exhibit high microwave-absorbing efficiency over wide frequency, with a minimum reflection loss *(RL*_min_) of −59.40 dB and an absorption bandwidth of 4.2 GHz [17]. Huang et al. fabricated multilayered elastomeric meta-structure with dielectric-magnetic nano lossy composites and patterned resistive films, covering −10 dB bandwidth of 5.74 GHz–18.00 GHz with an *RL*_min_ of −49 dB at 16.2 GHz [18]. However, it is still a significant challenge to obtain low-density elastomeric microwave absorbers with strong and broadband microwave absorption performances due to the random distribution of electromagnetic lossy blocks [19].

It is well-accepted that the desirable properties of elastomeric composites are not only associated with the type and loading of functional fillers but also their dispersion and distribution in the matrix [20]. Recently, versatile elastomeric materials with excellent mechanical and functional properties have been developed by applying the concept of double percolation, where fillers are selectively distributed and percolated through one of the phases or at the interface of a co-continuous binary blend [21,22,23]. Compared with single-phase polymer composites, double percolation demonstrates greater improvements in mechanical, charge storage, electrical, and magnetic properties at lower filler concentrations [24,25]. For instance, selective localization of multi-walled carbon nanotubes in polypropylene/natural rubber blends reduces the rheological percolation threshold by 2 wt.% MWCNTs [26]. Acrylonitrile-butadiene-styrene with selectively located and homogeneously dispersed graphene in the styrene-acrylonitrile phase shows an extremely low electrical percolation threshold of 0.13 vol.% [27]. Our previous work in natural rubber/epoxidized natural rubber (NR/ENR) with the selective distribution of conductive carbon black also confirmed the double percolation enhanced microwave attenuation, which can be tuned by compatibility and blend ratio of NR/ENR [28,29,30]. However, the microwave-absorbing performance of the as-prepared composites still cannot catch up with the requirements of practical applications. Moreover, owing to more complex components and structures, the dual percolation-reinforced microwave attenuation mechanism is complicated and still not so clear.

Compared with wet-based compounding technologies, including solution compounding, latex compounding, and in situ polymerization, mechanical compounding is considered the most effective protocol for large-scale manufacturing of elastomeric MAMs. In our previous work, using a two-roll blending method, the electromagnetic performances of NR composites were significantly improved by exploiting the synergetic effect of MWCNTs and MoS_2_ [31]. Herein, to obtain a balanced electromagnetic parameter for better microwave attenuation, natural rubber/nitrile butadiene rubber (NR/NBR) with confined distributions of multi-walled carbon nanotubes (MWCNTs) and molybdenum disulfide (MoS_2_) was fabricated via a compartmentalized approach. Compared with direct compounding, masterbatch-based compounding exhibits better-controlled distributions of electromagnetic blocks and subsequently stronger microwave-absorbing performances. Particularly, elastomeric composites with MWCNTs in the NR phase and MoS_2_ in the NBR phase demonstrated the best microwave attenuation capacity, with an *RL*_min_ of −44.54 dB and a broad absorption bandwidth up to 3.60 GHz. Moreover, the evolution fabrication-tailored morphology and its associated electromagnetic loss mechanism were discussed.

## 2. Materials and Methods

### 2.1. Materials

Natural rubber (NR, density 0.93 g/cm^3^) was provided by Hainan Natural Rubber Industry Group Co., Ltd., Haikou, China; nitrile butadiene rubber (NBR, density 0.97 g/cm^3^) was purchased from the Qilu Branch of Sinopec, Zibo, China; multi-walled carbon nanotubes (MWCNTs, density 2.10 g/cm^3^) were purchased from Nanjing Xianfeng Nanomaterials Technology Co., Ltd., Nanjing, China; molybdenum disulfide (MoS_2_, density 4.80 g/cm^3^) was purchased from Guangzhou Nano Chemical Technology Co., Ltd., Guangzhou, China; other reagents such as sulfur (S), zinc oxide (ZnO), stearic acid (SA), and N-cyclohexyl-2-benzothiazole sulfonamide (CBS) were of analytical grade and used without further purification.

### 2.2. Preparation of Composites

All composites with the components listed in Table 1 were prepared by mechanical compounding around 60 ± 5 °C (Figure 1) and subsequent vulcanization. For a better comparison, all the filler content was fixed at 10 vol.% (volume fraction) of the matrix, and the volume ratio of MWCNTs to MoS_2_ was 1:1. Before compounding, raw rubber was homogenized 8 times on a two-roll mill.

Direct compounding: Initially, an appropriate amount of matrix (NR or/and NBR) was masticated in a min internal mixer for 6 min. After that, premixed MWCNTs and MoS_2_ were added and mixed for 6 min; then, the vulcanization ingredient was added and mixed for another 6 min. Once the compounds were mixed, gummers were obtained by laminating the compound on a two-roll mill 6 times. After being stored at room temperature for 24 h, the laminated compounds were compression vulcanized (145 °C, 15 MPa) in a desired mold for their optimum cure times deduced from the moving die rheometer (MDR 2000, Alpha, Hudson, OH, USA). According to the different components, as-prepared samples were designated as MWCNTs-MoS_2_/NR, MWCNTs-MoS_2_/NBR, and MWCNTs-MoS_2_/NR-NBR, respectively.

Masterbatch compounding: Initially, an appropriate amount of NR was masticated in a min internal mixer for 6 min. Then, MWCNTs or MoS_2_ was added and mixed for 3 min, forming NR masterbatches with only MWCNTs or MoS_2_. NBR masterbatch with only MWCNTs or MoS_2_ was prepped following the same procedure. After that, halves of different masterbatches were pairwise combined and mixed for another 3 min. After being mixed with vulcanization ingredients for another 6 min, the compounds were laminated on a two-roll mill 6 times, resulting in flat gummers. After being stored at room temperature for 24 h, the laminated compounds were compression vulcanized (145 °C, 15 MPa) in a desired mold for their optimum cure times deduced from the moving die rheometer (MDR 2000, Alpha, Hudson, OH, USA). According to the different components, as-prepared samples were designated MWCNTs-NR/MoS_2_-NR, MWCNTs-NBR/MoS_2_-NBR, MWCNTs-NBR/MoS_2_-NR, and MWCNTs-NR/MoS_2_-NBR, respectively.

### 2.3. Characterization

**XRD**. D8 advanced diffractometer analysis was performed with the X-ray diffraction (XRD, Rigaku, Japan) of the sample by a Cu-Kα source (λ = 0.154 nm) at a scanning rate of 1/min.

**Raman**. A Raman spectroscope was investigated with the spectral range of 0 cm^−1^ − 4000 cm^−1^ by a Lab RAM HR Evolution Raman imaging microscope (Palaiseau, France). 

**SEM&EDS** The micromorphology of the composite was analyzed by an EDAX energy spectrum-integrated Hitachi S-4800 scanning electron microscope (Tokyo, Japan). Before visualization, all rubber composites were fractured at low temperatures to obtain a cross profile and coated with platinum.

**Electromagnetic parameters**. The electromagnetic parameters, i.e., relative complex permittivity and relative complex permeability, of the as-fabricated microwave-absorbing material over 2.00–18.00 GHz were obtained via the coaxial transmission line method using an Agilent N5244A vector network analyzer (Santa Clara, CA, USA). The composite material was vulcanized into a coaxial sample with an outer diameter of 7.00 mm, an inner diameter of 3.00 mm, and a thickness of 2.00 mm.

## 3. Results and Discussion

### 3.1. Nanostructure of Fillers

The characteristic of filler is an essential factor in determining the wave absorption performance of composite materials, so the crystal structure of filler was analyzed. Figure 2a is the XRD spectrum of MWCNTs and MoS_2_. It can be observed that MWCNTs show obvious diffraction peaks at 25.57° and 43.13°, corresponding to their (002) and (100) crystal plane diffraction characteristic peaks. The sharp and narrow peaks at 14.44°, 29.07°, 32.75°, 33.57°, 35.95°, 39.62°, 44.24°, 49.88°, 56.10°, 58.42° and 60.31° in MoS_2_ correspond to the diffraction characteristic peaks of the (002), (004), (100), (101), (102), (103), (006), (105), (106), (110), and (008) crystal planes, respectively. This is consistent with the diffraction pattern of the 2H-MoS_2_ standard card (JCPDS No. 37-1492), indicating the high crystallinity of MoS_2_. Figure 2b reveals the Raman spectra of MWCNTs and MoS_2_. It can be seen that MWCNTs exhibit two obvious peaks at 1334.25 cm^−1^ and 1572.56 cm^−1^, which are ascribed to the D-band characteristic peak caused by sp^2^ hybrid carbon defect and dislocation and the G-band characteristic peak stemmed from higher order E_2g_ mode of graphitized layers, respectively. Correspondingly, MoS_2_ has two characteristic peaks at 314.34 cm^−1^ and 376.24 cm^−1^, which are attributed to its in-plane and out-of-plane vibrations in turn [32]. According to the Prague theory, by the wave number difference (~25.5 cm^−1^) corresponding to the two vibration peaks, it can be calculated that the number of stacked MoS_2_ layers is 7–70 [33]. Furthermore, we characterized the micromorphology of MWCNTs and MoS_2_ (Appendix A), where MWCNTs powder shows aggregates of MWCNTs with a diameter of about 50 nm. As shown in Appendix A, MoS_2_ is made of multiple layers with a thickness of about 4.5~53.4 nm. According to the thickness of single-layer MoS_2_ reported in the literature (~0.65 nm) [34], it can be estimated that there are about 7~80 layers stacked, which consists of the data obtained by XRD.

### 3.2. Morphology of Composites

Generally, the wave absorption of rubber composite is derived from the electromagnetic loss network throughout the matrix. To explore the influences of blending processes on the microstructure of the composite, the cross-section of the composite was visualized by SEM (Figure 3). It can be observed that cross-sections of all composites are relatively rough, with obvious MWCNTs and MoS_2_ promontories. Because NBR is a polar molecule with poor compatibility with MWCNTs or MoS_2_, there are many aggregates in MWCNTs-MoS_2_/NBR (Figure 3b) and MWCNTs-NBR/MoS_2_-NBR (Figure 3b′). Compared with MWCNTs-MoS_2_/NBR, the accumulation of filler in NR composite is smaller, and the dispersion is better (Figure 3a,a′). Furthermore, the size of MWCNTs and MoS_2_ aggregates further decrease in composites with both NR and NBR, indicating that the combination of NR and NBR is conducive to dispersing and forming a better electromagnetic loss network (Figure 3c). Compared with the direct compounding method, the distribution of MWCNTs and MoS_2_ in the masterbatch-compounded composite is the best, with more intact electromagnetic loss networks of MWCNTs and MoS_2_ (Figure 3c′). Owing to the high viscosity of the matrix, MWCNTs or MoS_2_ in masterbatch is difficult to migrate to another phase, forming an interpenetrating network structure with the selective distribution of MWCNTs or MoS_2_ in binary rubber blend [35]. Such hierarchical structure with multiple electromagnetic loss networks and abundant interfaces is conducive to electromagnetic wave attenuation of materials. As both matrices involve vulcanization-induced irreversible chemical crosslinking, it is challenging to distinguish the NR phase and NBR phase in composites by selective etching. However, evidenced by the Mo elements mapping of all the composites (Appendix A), the MoS_2_ network structure of the composite obtained by masterbatch compounding is better than that of their counterparts obtained by direct compounding. Interestingly, when the components in the master batch are exchanged, the electromagnetic loss networks in the MWCNTs-NBR/MoS_2_-NR composite become worse (Appendix A), which may be due to the poor compatibility of MWCNTs with NBR and MoS_2_ with NR.

### 3.3. Microwave-Absorbing Properties of Composites

The microwave absorption performance of a microwave absorber is usually assessed by its reflection loss (*RL*), which is related to internal characteristics (such as complex permittivity, complex permeability, etc.) and external use conditions (such as thickness, operating frequency, etc.). *RL* can be simulated by the transmission line theory as the following formula [36]:(1)$$RL=20lg\left|\frac{Z_{\mathrm{in}}-Z_{0}}{Z_{\mathrm{in}}+Z_{0}}\right|,$$
(2)$$Z_{\mathrm{in}}=Z_{0}\sqrt{\frac{\mu_{r}}{\varepsilon_{r}}}\tanh\left(j\frac{2\pi fd}{c}\sqrt{\mu_{r}\varepsilon_{r}}\right),$$
where *ε*_r_ is the relative complex permittivity, *μ*_r_ is the relative complex permeability, *f* is the frequency of the electromagnetic wave (Hz), *d* is the thickness of absorbing material (m), and *c* is the velocity of the electromagnetic wave in free space (m/s). To evaluate the microwave-absorbing performances, the *RL*s of each composite are calculated at different thicknesses ranging from 1.0 to 5.5 mm with a step of 0.1 mm. Figure 4 presents two-dimensional contour maps of *RL*s of composite materials as a function of thickness and frequency. It can be seen that calculated *RL*s of as-prepared composites vary with the frequency of the electromagnetic wave and the thicknesses of the absorbers, implying irregularity of the electromagnetic loss network in the composite. It is worth noting that the match frequency corresponding to *RL*_min_ shifts to a lower frequency with the increasing thickness of the absorber, which is consistent with the quarter-wavelength match principle and related to the impedance matching involved in the waveguide transmission line theory [37]. Owing to the better dispersion of MWCNTs and MoS_2_ in NR, the *RL*_min_ of MWCNTs-MoS_2_/NR is −24.53 dB (16.56 GHz), which is better than that of MWCNTs-MoS_2_/NBR (−15.92 dB, 12.32 GHz). The combination of NR and NBR further enhances the microwave-absorbing performance, confirmed by the *RL*_min_ of −31.24 dB for MWCNTs-MoS_2_/NR-NBR at the frequency of 8.64 GHz and the thickness of 2.4 mm. Such enhanced microwave attenuation capacity of MWCNTs-MoS_2_/NR-NBR not only results from more heterogeneous interfacial of NR-NBR but also thermodynamics favored preferential location of MWCNTs and MoS_2_ in co-continuous NR/NBR blend [38]. Overall, composites fabricated via masterbatch compounding show similar tendencies with stronger microwave-absorbing performances. The *RL*_min_ of MWCNTs-NR/MoS_2_-NR, MWCNTs-NBR/MoS_2_-NBR, and MWCNTs NR/MoS_2_-NBR reaches −30.07 dB, −29.03 dB, and −44.54 dB, which is 1.23 times, 1.82 times, and 1.43 times of that of MWCNTs-MoS_2_/NR, MWCNTs-MoS_2_/NBR, and MWCNTs-MoS_2_/NR-NBR, respectively. In addition to being as low *RL* as possible, the effective absorption bandwidth (EAB, *RL* < −10 dB) should be as wide as possible for the practical application of MAMs. It can be seen that EABs of all composites are above 2.00 GHz, which should result from the better impedance matching derived from balanced electromagnetic parameters [35]. Particularly, the EAB of MWCNTs-NR/MoS_2_-NBR composite reaches 3.60 GHz, which is higher than that of our previous work [31]. Moreover, effective absorption of the incident electromagnetic waves ranging from 4.00 GHz to 18.00 GHz can be achieved by simply adjusting the thickness of the composite, demonstrating versatile broadband absorption characteristics. When the components in the masterbatch are exchanged to MWCNTs in NBR and MoS_2_ in NR, the *RL*_min_ and EAB shown in Appendix A deteriorate to −18.23 dB (17.84 GHz) and 2.40 GHz (5.28~5.52 GHz and 15.52~17.68 GHz), which is consistent with the results of SEM.

To further compare the influence of the blending process on the wave absorption performance of composite materials, the *RL* curves of as-prepared composites at the optimal thickness were extracted (Figure 5a). Composites fabricated via masterbatch compounding exhibit smaller thicknesses than ones via direct compounding. The optimized thickness of MWCNTs-NR/MoS_2_-NR, MWCNTs-NBR/MoS_2_-NBR, and MWCNTs-NR/MoS_2_-NBR is 3.70 mm, 2.40 mm, and 2.10 mm, only accounting for 88.10% of MWCNTs-MoS_2_/NR, 43.64% of MWCNTs-MoS_2_/NBR, and 87.50% of MWCNTs-MoS_2_/NR-NBR, respectively. Figure 5b depicts a comparison of some reported works on elastomeric MAMs in terms of *RL*_min_ and EAB [8,13,16,17,39,40,41]. Compared with other counterparts, MWCNTs-NR/MoS_2_-NBR in this work demonstrates many desired features, including lower density, higher absorption, broader bandwidth, and thinner absorption thickness, indicating apparent competitivity.

### 3.4. Microwave Attenuation Mechanism

Theoretically, the relative complex permittivity (*ε*_r_ = *ε*′ − *jε*″) and relative complex permeability (*μ*_r_ = *μ*′ − *jμ*″) of composites determine the performance of a microwave absorber. The real parts of the permittivity (*ε*′) and the permeability (*μ*′) represent the ability to store electrical energy and magnetic energy, while the imaginary parts of the permittivity (*ε*″) and the permeability (*μ*″) refer to the loss ability of electric and magnetic energy, respectively. Figure 6 presents the measured electromagnetic parameters as a function of frequency. It can be seen from Figure 6a–c that the real permittivity of the composite decreases with the increase in frequency, which should be attributed to the enhanced polarization lagging at a higher frequency [42] and the existence of the dispersion effect [43]. Similarly, *ε*″ values of all samples also exhibit a decreasing trend with a fluctuation around the 9.0 GHz range of 2–18 GHz, and this fluctuation is ascribed to the presence of multiple dielectric relaxation polarizations [44]. Due to the enhanced interface polarization from more heterogeneous interfaces between NR and NBR, composites with both NR and NBR show higher complex permittivity compared with composites with only NR or NBR. For example, the real part, imaginary part, and dielectric loss tangent (Tan *δ*_ε_ = *ε*″/*ε*′) of MWCNTs-MoS_2_/NR-NBR composites is 10.80~17.31, 0.97~5.83, and 0.09~0.34, while that of MWCNTs-MoS_2_/NR composites is only 8.04~12.61, 0.50~3.18, and 0.06~0.32, respectively. Moreover, compared with their counterparts from direct compounding, masterbatch compounded composites show higher *ε*′, *ε*″, and *δ*_ε_, indicating stronger dielectric loss capacities. As shown in Figure 6a′−c′, all values of *μ*′, *μ*″, and magnetic loss tangent (Tan *δ*_ε_=*μ*″/*μ*′) for composites demonstrate an increasing trend with slight fluctuations around 1.0, 0.0, and 0.0, which is probably caused by stronger multiple natural resonances at higher frequency [45]. In addition, values of complex permeability are much lower than that of the complex permittivity, implying that the microwave attenuation capacity of the as-prepared composites is dominated by the dielectric loss mechanism [17].

Impedance matching (*Z*) and attenuation constant (*α*) are the other two vital parameters affecting the MA properties of absorbers, which indicate the capability of allowing incident electromagnetic waves to penetrate inside of the absorber and dissipating electromagnetic energy in other forms, respectively [46]. *Z* and *α* can be calculated as follows:(3)$$Z=\left|\sqrt{\frac{\mu_{r}}{\varepsilon_{r}}}\right|,$$
(4)$$\alpha =\frac{\sqrt{2}\pi f}{c}\sqrt{\left(\mu^{\prime \prime }\varepsilon^{\prime \prime }-\mu^{\prime }\varepsilon^{\prime }\right)+\sqrt{\left({\mu^{\prime \prime }}^{2}+{\mu^{\prime }}^{2}\right)\left({\varepsilon^{\prime \prime }}^{2}+{\varepsilon^{\prime }}^{2}\right)}}.$$

There is no doubt that a desired microwave absorber should allow as much electromagnetic wave as possible to penetrate in and then convert them into another form of energy. Figure 7 displays values of *Z* and *α* of as-prepared composites. It can be observed from Figure 7a that composites with both NR and NBR via both methods show inferior impedance matching compared with their counterparts with sole NR or NBR, evidenced by their *Z* values that are farther away from 1. Similar tendencies can be seen for masterbatch compounded composites vs. direct compounded ones, which is attributed to the more intact conductive network stemming from the selective distribution of MWCNTs and MoS_2_ (Figure 3). On the contrary, NR/NBR-containing or masterbatch-compounded composites exhibit higher attenuation constant (Figure 7b), indicating stringer microwave-absorbing capacity. In detail, α values of MWCNTs-MoS_2_/NR-NBR via direct compounding is 29.26–207.37, which is higher than that of MWCNTs-MoS_2_/NR (15.48~137.96) or MWCNTs-MoS_2_/NBR (17.9~145.95). When masterbatch compounding is adopted, the attenuation constant MWCNTs-NR/MoS_2_-NBR further increases to 40.04~257.89, demonstrating the strongest microwave attenuation ability. Evidenced by the higher *ε*′, *ε*″, and *δ*_ε_ (Figure 6), a more intact interconnected network from masterbatch compounding endows the as-prepared composites with better electrical conductivity, which results in impedance matching but facilitates microwave attenuation via conductive loss stemming from micro-currents and capacitors [45].

As aforementioned, it can be deliberately inferred that synergetic effects of co-continuous binary blend-induced heterogeneous interfaces and fabrication-boosted selective distribution of electromagnetic loss blocks contribute to the enhanced microwave attenuation capacity of MWCNTs-NR/MoS_2_-NBR composite, which is schematically illustrated in Figure 8. Initially, compared with composites with sole NR or NBR, NR/NBR demonstrates more heterogeneous interfaces, which is conducive to attenuating more electromagnetic waves via stronger interfacial polarization. Then, the interfacial wettability difference between fillers and matrix favors the preferential location of MWCNTs and MoS_2_ in co-continuous NR/NBR, boosting the more intact electromagnetic loss network and subsequently higher conduction loss. Moreover, the hierarchical structure with dual-filler networks and the co-continuous blend is propitious to the multiple scattering/reflection of incident electromagnetic waves, prolonging the propagation of electromagnetic waves in the composite. Last but not least, the interfaces between MWCNTs and MoS_2_ are equivalent to a capacitor-like circuit due to the accumulation of charges, which can effectively attenuate incident microwaves. Therefore, this work paves the way to developing elastomeric microwave-absorbing materials with high absorption over a wide frequency range.

## 4. Conclusions

Natural rubber (NR)/nitrile butadiene rubber (NBR) blends (NR/NR) with confined distributions of multi-walled carbon nanotubes (MWCNTs) and molybdenum disulfide (MoS_2_) were prepared via mechanical blending by adopting a masterbatch compounding approach. Owing to interfacial wettability difference and fabrication-induced confinement, selective distribution of MWCNTs and MoS_2_ was achieved in the masterbatch-compounded composite. When MWCNTs were in the NR phase and MoS_2_ is in the NBR phase, the *RL*_min_ of the composite material was up to −44.54 dB, and the effective absorption bandwidth was as high as 3.60 GHz. Analysis of morphology-associated electromagnetic behaviors indicate that such enhanced microwave attenuation capacity of MWCNTs-NR/MoS_2_-NBR is stemmed from conduction loss, interfacial polarization, and multiple scattering reflections. This fabrication-tailored hierarchical structuring in the rubber blends can be adopted to design elastomeric microwave absorbers with high attenuation capacity over a wide frequency range, extending their applications in microwave anechoic chambers, electronic devices, etc.

## Figures and Tables

**Figure 1 nanomaterials-13-01644-f001:**
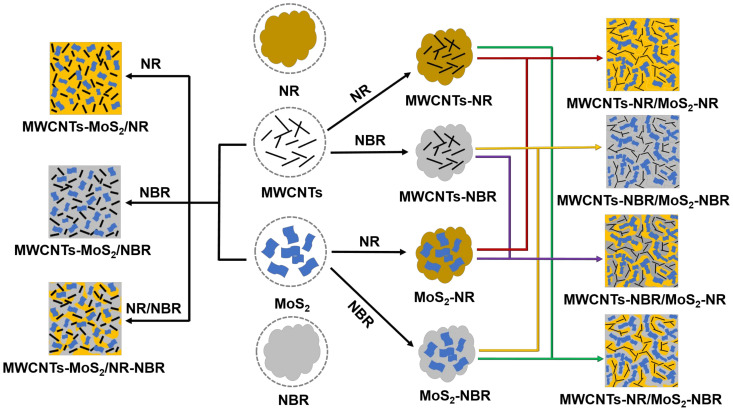
Procedure description of the fabrication of NR or/and NBR composites with MWCNTs and MoS_2_ and color lines indicate different combinations.

**Figure 2 nanomaterials-13-01644-f002:**
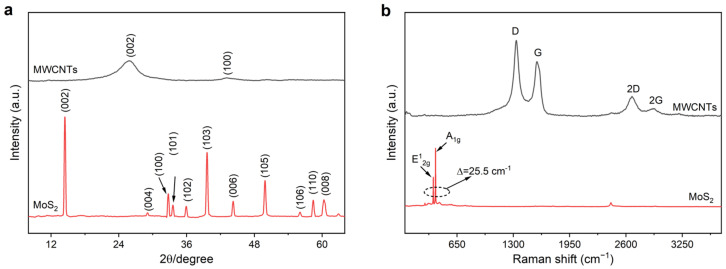
(**a**) XRD and (**b**) Raman spectra of MWCNTs and MoS_2_.

**Figure 3 nanomaterials-13-01644-f003:**
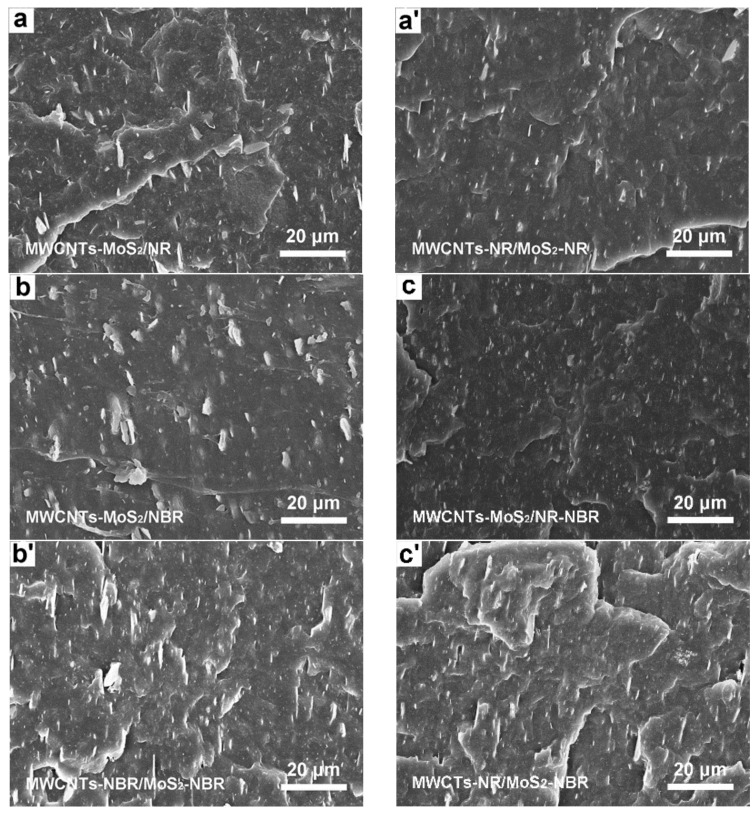
SEM images of as-prepared composites via different fabrication processes: (**a**–**c**) direct compounding and (**a**′–**c**′) masterbatch compounding.

**Figure 4 nanomaterials-13-01644-f004:**
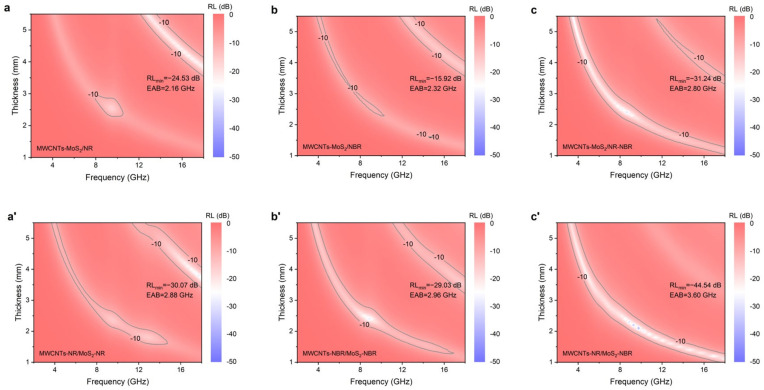
Contour maps of thickness-depended reflection loss over the frequency of 2–18 GHz for resulted composites: (**a**–**c**) direct compounding and (**a**′–**c**′) masterbatch compounding.

**Figure 5 nanomaterials-13-01644-f005:**
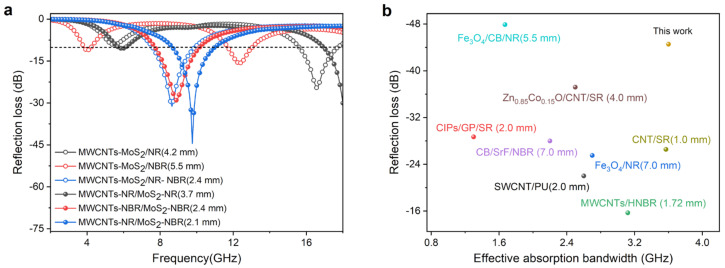
(**a**) *RL*s at the optimized thickness for resulted composites and (**b**) comparison of the microwave attenuation capacity of reported elastomeric materials.

**Figure 6 nanomaterials-13-01644-f006:**
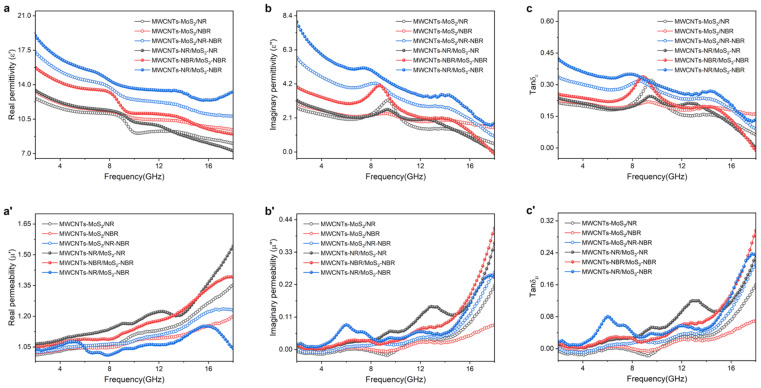
Frequency-depended (**a**–**c**) complex permittivity and (**a′**–**c′**) complex permeability of as-prepared composites.

**Figure 7 nanomaterials-13-01644-f007:**
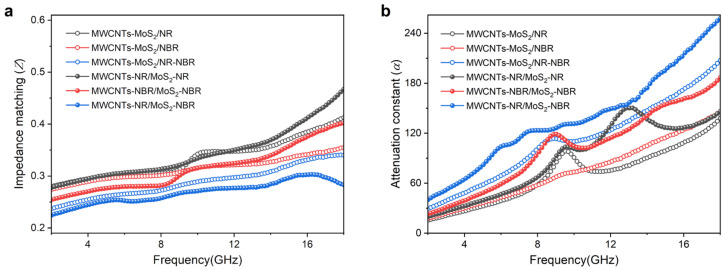
(**a**) Impedance matching and (**b**) attenuation constant of as-prepared composites.

**Figure 8 nanomaterials-13-01644-f008:**
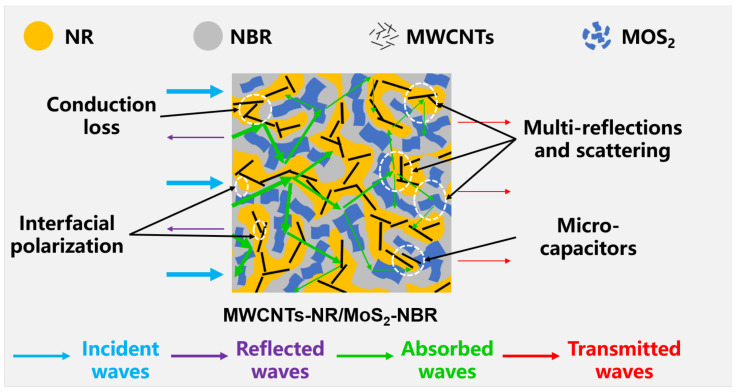
Schematic diagram of the microwave attenuation for MWCNTs-NR/MoS_2_-NBR composites.

**Table 1 nanomaterials-13-01644-t001:** Components of composites.

Component	NR/g	NBR/g	MWCNTs/g	MoS_2_/g	S/g	ZnO/g	SA/g	CBS/g
MWCNTs-MoS_2_/NR	40.00	0.00	4.20	9.60	0.60	1.00	0.40	0.56
MWCNTs-MoS_2_/NBR	0.00	40.00	4.20	9.60	0.60	1.00	0.40	0.56
MWCNTs-MoS_2_/NR-NBR	20.00	20.00	4.20	9.60	0.60	1.00	0.40	0.56
MWCNTs-NR/MoS_2_-NR	40.00	0.00	4.20	9.60	0.60	1.00	0.40	0.56
MWCNTs-NBR/MoS_2_-NBR	0.00	40.00	4.20	9.60	0.60	1.00	0.40	0.56
MWCNTs-NBR/MoS_2_-NR	20.00	20.00	4.20	9.60	0.60	1.00	0.40	0.56
MWCNTs-NR/MoS_2_-NBR	20.00	20.00	4.20	9.60	0.60	1.00	0.40	0.56

## Data Availability

Not applicable.

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
