# Peer review of "Morphology and Microwave-Absorbing Performances of Rubber Blends with Multi-Walled Carbon Nanotubes and Molybdenum Disulfide"

_nanomaterials, 2023, doi:10.3390/nano13101644_

Round 1

Reviewer 1 Report

The electromagnetic performances of natural rubber and multi-walled carbon nanotubes (MWCNTs) and molybdenum disulfide (MoS2) composites were synthesized and investigated.  In order to obtain a balanced electromagnetic parameter for better microwave shielding, natural rubber/nitrile butadiene rubber with confined distributions of MWCNTs and MoS2 was fabricated. Natural rubber/nitrile butadiene rubber blends with confined distributions of MWCNTs and MoS2 were prepared via mechanical blending. Analysis of morphology-associated electromagnetic behaviors indicates that such enhanced microwave attenuation capacity is provided by conduction loss, interfacial polarization, and multiple scattering reflections. As compared with other absorbers, the composites in this work have lower density, higher absorption, broader bandwidth, and thinner absorption thickness. So, the proposed structures can be adopted to design elastomeric microwave absorbers with high attenuation capacity over a wide frequency range. I have few comments to the paper.

1. Please, give some comments to the increase of the real part of magnetic permittivity with increase of frequency in Figure 6 a′. What is the physical reason for the maximums near 8-9 GHz in the imaginary part of complex permittivity for most of the compositions in Figure 6b?

2. In p.6 line 221, the words “dielectric property” should be replaced by “dielectric permittivity”.

3. In p.9 lines 321 – 325 and Figure 7, the attenuation constant α values must be given with measurement units.

No comments.

Author Response

Dear Reviewer,

Thank you very much for the comments concerning our manuscript, “Morphology and Microwave-absorbing Performances of Rubber Blends with Multi-walled Carbon Nanotubes and Molybdenum Disulfide”. These comments are valuable and helpful for revising and improving our paper. We have carefully checked the manuscript and made many corrections. We hope it meets the requirements for the publication of nanomaterials. All revisions to the manuscript were blue-marked, and our point-by-point responses to the comments are as follows:

Responds to comments:

Q1. Please, give some comments to the increase of the real part of magnetic permittivity with increase of frequency in Figure 6 a. What is the physical reason for the maximums near 8-9 GHz in the imaginary part of complex permittivity for most of the compositions in Figure 6b?

Response: More information has been provided, please see lines 287-290, lines 295-297, and line 303 in the revised manuscript.

Q2. In p.6 line 221, the words “dielectric property” should be replaced by “dielectric permittivity”.

Response: Thanks for your suggestion, and the “dielectric property” has been replaced in the revised manuscript.

Q3. In p.9 lines 321 – 325 and Figure 7, the attenuation constant α values must be given with measurement units.

Response: Generally, dB/m is used as a unit for the attenuation constant in some engineering filed. However, we think the attenuation constant used in our manuscript is a dimensionless parameter due to the different calculation methods, which can be found in many references such as Carbon, 2019,143, 507-516; Journal of Materials Chemistry A, 2022,10, 8479-8490.

Again, thank you very much for the comments and suggestions. Please contact us if there are any problems or questions about our manuscript.

Yours sincerely,

Yanfang Zhao

Reviewer 2 Report

In general, the manuscript makes a positive impression. The very idea of these studies to separate and then sum up the functions of fillers in a composite (“rod” (carbon nanotubes) and lamellar (MoS2) filler) to obtain a final material with desired properties is very useful and interesting.

There are a few small remarks.

1.       Figure S2  shows not Element mapping of Mo for resulting composites, but black squares.

2.       The authors do not discuss the reasons why the dependencies presented in Figs. 6 and 7 are different. As a consequence, the proposed by the authors the scheme (Fig.8) does not seem to be confirmed by experiment.

Author Response

Dear Reviewer,

Thank you very much for the comments concerning our manuscript titled Morphology and Microwave-absorbing Performances of Rubber Blends with Multi-walled Carbon Nanotubes and Molybdenum Disulfide. All these comments are valuable and helpful for revising and improving our paper. We have carefully checked the manuscript and made many corrections. We hope it meets the requirements for the publication of nanomaterials. All revisions to the manuscript were blue-marked, and our point-by-point responses to the comments are as follows:

Responds to comments:

Q1: Figure S2 shows not Element mapping of Mo for resulting composites, but black squares.

Response: Thank you for your comment. This problem might result from Figures with too small dimensions. Figures with larger dimensions were provided in the revised manuscript, where you can find there are many pink pixels with a back background.

Q2. The authors do not discuss the reasons why the dependencies presented in Figs. 6 and 7 are different. As a consequence, the proposed by the authors the scheme (Fig. 8) does not seem to be confirmed by experiment.

Response: More information has been provided, please see lines 287-290, lines 295-300, and lines 327-331 in the revised manuscript.

Again, thank you very much for the comments and suggestions. If there are any problems or questions about our manuscript, please contact us.

Yours sincerely,

Round 2

Reviewer 1 Report

The revised paper is suitable to be published.